# DRL-STNet: Unsupervised Domain Adaptation for Cross-modality Medical Image Segmentation via Disentangled Representation Learning

Hui Lin[0000-0002-6559-2751], Florian Schiffers[0000-0003-3959-5163],
Santiago López-Tapia[0000-0003-2090-7446], Neda Tavakoli[0000-0002-1541-5917],
Daniel Kim[0000-0003-2660-8973], Aggelos K. Katsaggelos[0000-0003-4554-0070]

Northwestern University, Evanston, IL 60208, USA
Corresponding author: huilin2023@u.northwestern.edu

**Abstract.** Unsupervised domain adaptation (UDA) is essential for medical image segmentation, especially in cross-modality data scenarios. UDA aims to transfer knowledge from a labeled source domain to an unlabeled target domain, thereby reducing the dependency on extensive manual annotations. This paper presents DRL-STNet, a novel framework for cross-modality medical image segmentation that leverages generative adversarial networks (GANs), disentangled representation learning (DRL), and self-training (ST). Our method leverages DRL within a GAN to translate images from the source to the target modality. Then, the segmentation model is initially trained with these translated images and corresponding source labels and then fine-tuned iteratively using a combination of synthetic and real images with pseudo-labels and real labels. The proposed framework exhibits superior performance in abdominal organ segmentation on the FLARE challenge dataset, surpassing state-of-the-art methods by **11.4%** in the Dice similarity coefficient and by **13.1%** in the Normalized Surface Dice metric, achieving scores of 74.21% and 80.69%, respectively. The average running time is 41 seconds, and the area under the GPU memory-time curve is 11,292 MB. These results indicate the potential of DRL-STNet for enhancing cross-modality medical image segmentation tasks.

**Keywords:** Unsupervised domain adaptation · Organ segmentation· Cross-modality· Feature disentanglement· Self-training

## 1  Introduction

In the realm of medical imaging, accurate segmentation of anatomical structures is crucial for diagnostics, treatment planning, and patient monitoring [21,20,22]. However, acquiring annotated data for every imaging modality is both costly and time-consuming. This challenge is exacerbated when multiple modalities are involved, as it is impractical to obtain paired data for every patient due to logistical constraints.

Unsupervised domain adaptation (UDA) has emerged as a promising solution to address this issue, which can efficiently adapt models between modalities without the need for paired data [3,16,41,39,32,33]. Jiang et al. [16] and Yao et al. [41] applied disentangled representation learning for abdominal organ segmentation. However, the generalizability of their method to segment a wider range of abdominal organs across multiple sources and sequences remains uncertain. Recent studies have further improved the robustness and accuracy of these frameworks through variational approximation and self-training techniques [39,32,19]. Additionally, Shin et al. [33] incorporated transformers with GANs to learn intra- and inter-slice self-attentive image translation for continuous segmentation in the slice direction.

This paper presents DRL-STNet[1], a novel framework for cross-modality medical image segmentation that leverages generative adversarial networks (GANs) [8], disentangled representation learning [1], and self-training [17]. It involves two main steps as illustrated in Fig. 1: First, a source-to-target unpaired image translation model is trained using a disentangled GAN. This model generates synthetic images in the target modality while preserving the anatomical structures from the source modality. Second, a segmentation model is initially trained using labeled synthetic images and iteratively fine-tuned using a combination of synthetic and real images with pseudo-labels and real labels. DRL-STNet enables precise segmentation in the target modality without requiring annotations for target images or paired target-source domains. Our contributions are as follows:

- We introduce DRL-STNet, a novel unsupervised domain adaptation framework for cross-modality medical image segmentation.
- Disentangled representation learning effectively translates images between modalities while preserving anatomical structures without requiring paired data.
- Self-training via pseudo-labeling facilitates iterative improvements by incorporating unlabeled data into the segmentation process.
- We provide a comprehensive evaluation of DRL-STNet, demonstrating its robustness and accuracy across various abdominal organs, imaging sequences, and institutions, thereby highlighting its potential for clinical applications. It surpasses state-of-the-art methods on the FLARE challenge dataset, improving the Dice similarity coefficient by 11.4% and Normalized Surface Dice by 13.1%.

## 2   Method

In this study, annotations $Y^a$ for volume $X^a$ from the source modality $a$ (e.g., CT scans) are available, while annotations for the volume $X^b$ from the target modality $b$ (e.g., MRI scans) are not available. The goal is to achieve precise segmentation on the target volume, a common challenge in clinical applications

---

[1] The code is available at https://github.com/HuiLin0220/DRL-STNet.git

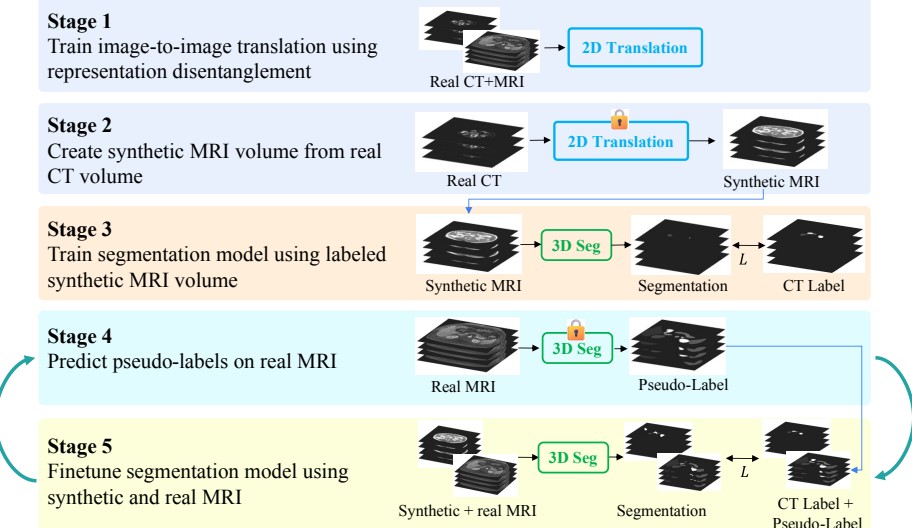

**Fig. 1.** Overview of the proposed DRL-STNet framework. The framework consists of five stages: **Stage 1-2:** Perform image translation from source to target. Train an image translation model based on disentangled representation learning to generate synthetic target volumes from real source volumes. **Stage 3-5:** Perform self-training via pseudo-labeling. Train the segmentation model using the synthetic target volumes and the corresponding source labels. Predict pseudo-labels on unlabeled target volumes and finetune the segmentation model with the combined data. Stages 4 and 5 are performed iteratively. The detailed architecture of the image translation model is described in Fig. 2. Viewing this figure in color is advised in the printed edition.

where obtaining annotations can be time-consuming, costly, or logistically difficult. Additionally, acquiring multiple imaging modalities from the same patient can be challenging due to logistical constraints. Even when possible, the process can take place several days apart, and the data may not be aligned. These factors often result in unpaired datasets $X^a$ and $X^b$. This study addresses these issues by focusing on unsupervised domain adaptation (UDA) to improve cross-modality segmentation accuracy. While this paper specifically addresses CT-to-MRI translation, the methodology can be applied to other modality pairs, such as PET-to-CT or ultrasound-to-MRI, depending on the specific clinical requirements and data availability.

As shown in Fig. 1, the proposed DRL-STNet framework addresses unsupervised domain adaptation (UDA) for cross-modality segmentation through image translation and segmentation. The framework includes a 2D image translation network that converts slices from the source modality $X^a$ to the target modality $X^b$. This allows us to synthesize an artificial target dataset (e.g., MRI) from the source data (CT) with ground-truth segmentation labels, enabling the train-

ing of a segmentation model that works in the MRI domain. While training a segmentation model

In the following section, we detail the network architectures for image translation and image segmentation.

### 2.1   Image Translation

Jiang et al. [16] and Yao et al. [41] have shown that disentangled learning is highly effective in style transfer, particularly for cross-modality medical imaging, while maintaining anatomical content. Inspired by them, our image translation model is composed of one shared content encoder $E_C$, two style encoders $E_S^a$ and $E_S^b$, one shared decoder $G$, and two image discriminators $D_a$ and $D_b$, and one content discriminator $D_c$, as depicted in Fig. 2.

All encoders and the decoder are based on ResNet [9], and all discriminators are based on LSGAN [30]. The image in each domain is disentangled into the content and style representations [18], $c^a$, $s^a$, $c^b$, and $s^b$. Each representation is a feature map obtained from the content or style encoder with a size of $C \times H \times W$ ($128 \times 128 \times 128$ in the following experiments), where $C, H, W$, respectively, represents the channel number, height, and width. The decoder $G$ reconstructs images by combining the content and style representations, obtaining four reconstructed images of the form $x^{ij} = G(c^i, s^j)$, where $i, j \in \{a, b\}$. Note that the same decoder $G$ is used for all image modalities, enabling it to learn a joint image representation. The discriminators $D_a$, $D_b$ are designed to distinguish at the image level, and $D_c$ are designed for the content level. A total of seven models, $\{E_C, E_S^a, E_S^b, G, D_a, D_b, D_c\}$, are jointly trained using the reconstruction and adversarial losses. The details about the losses are described in the following:

**Reconstruction loss:** Reconstruction losses at the image level are introduced to ensure the content and style encoders capture the entire image representation. The content representation $c^a$ ($c^b$) should contain all content information, and the style representation $s^a$ ($s^b$) should contain all style information in the modality $a$ ($b$). Based on this, the network should restore the original $x^a$ ($x^b$) from $c^a$ ($c^b$) and $s^a$ ($s^b$), constrained by:

$$\mathcal{L}_{rec} = \mathbb{E}_{x^a \in \chi^a} \left\| x^a - G(c^a, s^a) \right\| + \mathbb{E}_{x^b \in \chi^b} \left\| x^b - G(c^b, s^b) \right\|.$$

**Adversarial loss:** Adversarial losses at the image and content levels are used to maintain the image and feature alignment. In a generative adversarial network (GAN), the generator is trained to synthesize images to fool the discriminator, while the discriminator is trained to distinguish fake images from real ones. To ensure the quality of transferred images $x^{ba}$ ($x^{ab}$), $D_a$ ($D_b$) is trained to maximize $\mathcal{L}_{adv}^a$ ($\mathcal{L}_{adv}^b$), while $E_S^a, E_C, G$ ($E_S^b, E_C, G$) are trained to minimize $\mathcal{L}_{adv}^a$ ($\mathcal{L}_{adv}^b$). Additionally, $D_c$ is introduced to align content representation $\mathcal{L}_{adv}^c$.

The adversarial losses are defined as:

$$\mathcal{L}_{adv}^i = \mathbb{E}_{x^i \in \chi^i}[\log(D_i(x^i))] + \mathbb{E}_{c^j \in C^j, s^i \in S^i}[\log(1 - D_i(G(c^j, s^i)))]$$

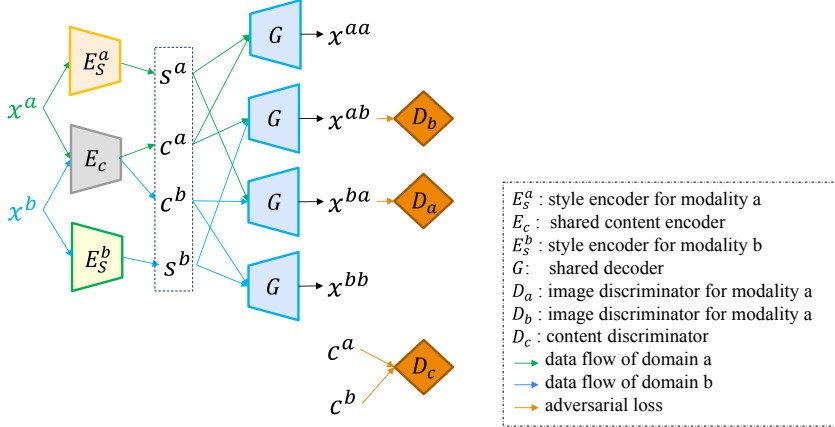

**Fig. 2.** The proposed image translation model using representation disentanglement. The model is composed of one shared content encoder $E_c$, two style encoders $E_s^a$ and $E_s^b$, and one shared decoder $G$. The image in each domain is disentangled into the content and style representations. The source image $(x^a)$ can be transferred into the target style $(b)$ by combining $c^a$ and $s^b$.

for $i, j \in \{a, b\}$ and $i \neq j$,

$$\mathcal{L}_{adv}^c = \mathbb{E}_{c^b \in C^b}[\log(D_c(c^b))] + \mathbb{E}_{c^a \in C^a}[\log(1 - D_c(c^a))].$$

The total adversarial loss is decomposed of the individual adversarial losses:

$$\mathcal{L}_{adv} = \mathcal{L}_{adv}^a + \mathcal{L}_{adv}^b + \mathcal{L}_{adv}^c.$$

Finally, the overall loss function follows the conventional min-max optimization known from GAN literature. To be minimized for the encoders and generator and maximized for the discriminators is:

$$\min_{(E_S^a, E_S^b, E_C, G)} \max_{(D_a, D_b, D_c)} \mathcal{L}(E_S^a, E_S^b, E_C, G, D_a, D_b, D_c) = \mathcal{L}_{adv} + \mathcal{L}_{rec}.$$

## 2.2    Self-Training via Pseudo-Labeling

In Stage 2, given a volume and its corresponding annotation $(X^a, Y^a)$ from the source domain, a slice $x^b$ from a volume in the target domain is randomly selected for the style representation. The $X^{ab}$ is generated through the 2D image translation model mentioned in Section 2.1 slice by slice. In Stage 3, the synthetic pairs $\{X^{ab}, Y\}$ are used to train a segmentation network $f$ that minimizes the segmentation loss:

$$\mathcal{L} = \sum \mathcal{L}_{seg}(Y^a, f(X^{ab}))$$

Then in Stage 4, the pseudo label $\hat{Y}^b$ of an unlabeled target scan $X^b$ is obtained by the trained segmentation model:

$$\hat{Y}^b = f(X^b)$$

Synthetic target scans may have distribution gaps compared to real target scans but come with precise annotations. In contrast, real target scans are paired with incomplete pseudo labels. Literature [33] shows that integrating labeled synthetic source scans $(X^{ab}, Y^a)$ and pseudo-labeled real target scans $(X^b, \hat{Y}^b)$ enhances the generalization ability. Therefore, these are combined in Stage 5 to fine-tune the previously trained segmentation model $f$ to minimize:

$$\mathcal{L} = \sum \mathcal{L}_{seg}(Y^a, f(X^{ab})) + \sum \mathcal{L}_{seg}(\hat{Y}^b, f(X^b))$$

A 3D self-configured nnU-Net [13] was utilized in this work for medical image segmentation to better capture the correlations among slices within a single scan. we do not optimize the segmentation efficiency.

### 2.3   Network Architecture

**Image Translation Network.** Our image translation module adopts an encoder-decoder architecture based on **ResNet** blocks, following the design principles of cycle-consistent adversarial networks. The architecture consists of:

- **Encoder:** Composed of several convolutional layers with residual connections, following the standard **ResNet** [9] structure. It progressively down-samples the input image while preserving spatial features through residual pathways.
- **Decoder:** Mirrors the encoder with upsampling operations (e.g., transposed convolutions or nearest-neighbor upsampling followed by convolution) and optional skip connections. It reconstructs the translated image from the encoded latent features.
- **Discriminator:** All discriminators are implemented based on **LSGAN (Least Squares GAN)** [30], which stabilizes adversarial training by minimizing the Pearson $\chi^2$ divergence. The discriminator consists of a sequence of convolutional layers with LeakyReLU activations and outputs patch-wise predictions (PatchGAN).

**Segmentation Network.** For medical image segmentation, we utilize a **3D self-configuring nnU-Net** [13], which automatically adapts its architecture and training pipeline to the specific characteristics of the dataset. Built upon the classic U-Net structure with an encoder-decoder design and long-range skip connections that fuse multi-scale contextual information. The network employs instance normalization and residual connections to improve training stability and segmentation performance. A combination of Dice loss and cross-entropy loss is used to optimize segmentation performance. This architecture enables effective modeling of 3D spatial context and robust generalization across varying image resolutions and anatomical structures.

## 3   Experiments

### 3.1   Dataset

The training dataset is curated from more than 30 medical centers under the license permission, including TCIA [4], LiTS [2], MSD [34], KiTS [10,11], autoPET [7,6], AMOS [15], LLD-MMRI [23], TotalSegmentator [37], and AbdomenCT-1K [29], and past FLARE Challenges [26,27,28]. The training set includes 2050 abdomen CT scans and over 4000 MRI scans. The validation and testing sets include 110 and 300 MRI scans, respectively, which cover various MRI sequences, such as T1, T2, DWI, and so on. The organ annotation process used ITK-SNAP [42], nnU-Net [14], MedSAM [24], and Slicer Plugins [5,25].

The pseudo labels generated by the FLARE22 algorithms [12,36] were utilized in this study. Due to variability in imaging orientations among MRI scans, we specifically selected unlabeled axial-view MRIs. Scans depicting non-abdominal regions—such as the heart, shoulder, or legs—were excluded to ensure anatomical consistency, retaining only those containing abdominal organs. A total of 50 CT scans and 50 MRIs were randomly selected for training the image translation models, regardless of MRI sequence type, to develop a robust model capable of translating CT images into various types of MRI sequences.

During the self-training stage, pseudo-labels for all abdominal MRIs were used directly without any selection or refinement, enabling the model to fully exploit the available pseudo-labeled data.

### 3.2   Pipeline

The translation pipeline consists of three main steps: cropping and resizing, z-score normalization, and feeding the processed images into the image translation model. Similarly, the segmentation pipeline involves z-score normalization followed by inference using the trained segmentation model.

To account for variations in size and resolution across the dataset, each slice was cropped and resized to a uniform resolution of $512 \times 512$ pixels. During preprocessing, z-score normalization was applied to standardize intensity values. Additionally, a range of data augmentation techniques was employed to improve model robustness, including random rotations, scaling, Gaussian noise, Gaussian blur, brightness and contrast adjustments, and mirroring.

For segmentation, we adopted a standard 3D sliding window approach with overlapping patches, where predictions in the overlapping regions were averaged to produce the final segmentation. No postprocessing was applied in our pipeline.

### 3.3   Evaluation measures

The evaluation metrics encompass two accuracy measures—Dice Similarity Coefficient (DSC) and Normalized Surface Dice (NSD)—alongside two efficiency measures—running time and area under the GPU memory-time curve. These

metrics collectively contribute to the ranking computation. Furthermore, the running time and GPU memory consumption are considered within tolerances of 60 seconds and 4 GB, respectively.

### 3.4   Implementation details

**Environment settings**  The development environments and requirements are presented in Table 1. For all experiments in our work, we utilized a workstation equipped with a single NVIDIA Quadro RTX 8000 GPU with 48 GB of memory, an Intel(R) Xeon(R) Gold 6226R CPU, and running CentOS 7.9.

**Table 1.** Development environments and requirements.

| System | CentOs 7.9 |
| --- | --- |
| CPU | Intel(R) Xeon(R) Gold 6226R CPU@1.2GHz |
| GPU (number and type) | 2 NVIDIA Quadro RTX 8000 48G |
| CUDA version | 12.4 |
| Programming language | Python 3.8 |
| Deep learning framework | torch 1.7.0, torchvision 0.8.1 |
| Specific dependencies | nnU-Net |

**Training protocols**  The training protocols for our experiments are summarized in Table 2. We initialized the network using the Kaiming normal distribution and trained it with a batch size of 2, using 3D patches of size $48 \times 192 \times 192$. The model was trained for a total of 800 epochs, employing Stochastic Gradient Descent (SGD) as the optimizer with an initial learning rate of 0.01. The learning rate followed a polynomial decay schedule. The entire training process spanned 17 hours. For the loss function, we combined Dice loss with cross-entropy to optimize segmentation performance. The model contained 30.71 million parameters, and the computational cost was measured at 1297.09 giga floating-point operations per second (GFLOPs).

## 4   Results and discussion

### 4.1   Image Translation

Examples of source, target, and generative slices are shown in Fig. 3. Since the difference between $x^{CT}$ and $x^{CT->CT}$ and the difference between $x^{CT}$ and $x^{CT->MRI->CT}$ are hard to tell, the content and style representations extracted from the encoders can fully represent the slice image in the CT domain and the decoder $G$ effectively reconstruct the image from these disentangled representations. The same conclusion applies to the MRI domain. Based on the conclusions above, $x^{CT->MRI}$ and $x^{MRI->CT}$ are highly likely to be reliable. For a UDA problem, it is hard to evaluate the quality of $x^{CT->MRI}$ and $x^{MRI->CT}$ quantitatively, since they are unpaired.

**Table 2.** Training protocols.

| | |
|---|---|
| Network initialization | Kaiming normal distribution |
| Batch size | 2 |
| Patch size | 48×192×192 |
| Total epochs | 800 |
| Optimizer | SGD with Nesterov momentum ($\mu = 0.99$) |
| Initial learning rate (lr) | 0.01 |
| Lr decay schedule | Poly learning rate schedule |
| Training time | 17 hours |
| Loss function | Dice and cross-entropy |
| Number of model parameters | 30.71M |
| Number of flops | 1297.09G |

### 4.2 Segmentation Results

**Comparison with state-of-the-art methods and ablation study** To compare the efficiency of our method with state-of-the-art approaches, we trained the segmentation model using the same architecture on synthetic MRIs generated by different Unsupervised Domain Adaptation (UDA) methods. The segmentation results on MRIs from the FLARE dataset's validation set are presented in Table 3. UDA methods, including CycleGAN [43], SIFA [3], and our proposed DRL-STNet, significantly enhance segmentation accuracy.

Without UDA, the segmentation performance is significantly lower across all organs, with an average Dice score of 6.13% and NSD of 6.00%. Both Cycle-GAN and SIFA improve segmentation quality, with SIFA achieving better overall results—yielding an average Dice of 67.52% ($p < 0.01$) and NSD of 67.52% ($p < 0.01$). DRL-STNet further enhances performance, reaching an average Dice of 72.07% ($p < 0.01$) and NSD of 72.07% ($p < 0.01$). The best results are achieved when DRL-STNet is combined with self-training, attaining an average Dice of 74.21% and NSD of 80.69%. However, the improvement in Dice score over DRL-STNet without self-training is not statistically significant ($p = 0.35$). These findings highlight the effectiveness of the self-training strategy in boosting segmentation accuracy, particularly in terms of surface alignment, making our full method (DRL-STNet + ST) the most effective among the compared approaches.

In terms of computational efficiency, all four methods exhibit similar inference times and GPU usage. This is largely because, aside from SIFA, all methods employ the same segmentation model architecture and share identical preprocessing and postprocessing pipelines. As a result, the computational cost remains consistent across w/o UDA, CycleGAN, and our proposed method. Although our method with self-training incurs a slightly higher computational cost than SIFA, the difference is not statistically significant. More importantly, it achieves sub-

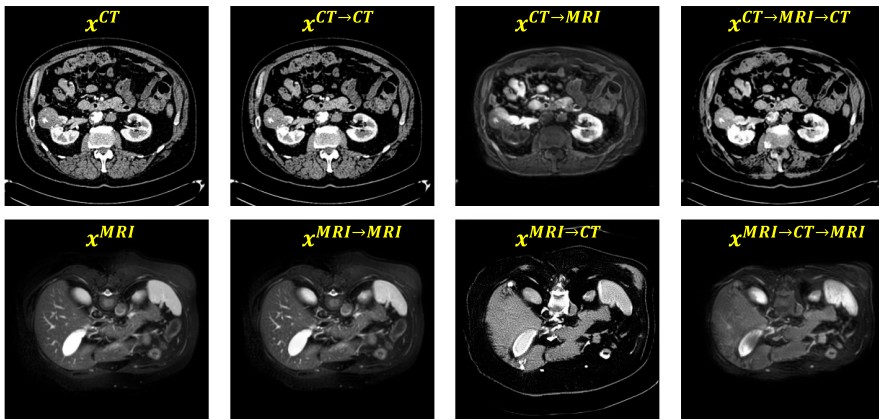

**Fig. 3.** Examples of source (CT), target (MRI), and generated slices produced by the proposed method. Since there is no ground truth for unpaired image translation, the small differences between the first and second columns, as well as between the first and fourth columns, suggest that our translation model is reliable.

stantially superior segmentation performance, offering a more favorable trade-off between accuracy and efficiency.

Notably, our method with self-training achieves the highest segmentation accuracy while maintaining an inference time of 34.7 seconds per scan and a GPU cost of 8.18 GB. This demonstrates the efficiency and practicality of our approach: it delivers state-of-the-art performance without compromising computational efficiency, making it highly suitable for real-world clinical deployment, where both accuracy and resource constraints are critical.

**Table 3.** Comparison between the state-of-the-art and the proposed method for cross-modality abdominal multi-organ segmentation on the FLARE dataset.

| Methods | DSC (%) ↑ | NSD (%) ↑ | Inference Time(s/scan) ↓ | GPU Cost(G) ↓ |
|---|---|---|---|---|
| w/o UDA | 6.40 ± 5.22 | 6.13 ± 5.87 | 32.9 ± 24.5 | 7.86 ± 3.99 |
| CycleGAN [43] | 44.98 ± 22.40 | 49.06 ± 12.81 | 35.7 ± 22.9 | 8.26 ± 4.14 |
| SIFA [3] | 62.81 ± 19.42 | 67.52 ± 13.99 | 28.1 ± 18.4 | 4.18 ± 2.40 |
| Ours w/o ST | 66.65 ± 17.52 | 72.07 ± 12.71 | 33.8 ± 26.7 | 8.04 ± 3.24 |
| **Ours** | **74.21 ± 16.21** | **80.69 ± 13.81** | 34.7 ± 29.5 | 8.18 ± 4.01 |

*Bold*: Best results; UDA: Unsupervised domain adaptation; ST: self-training
R.kid: Right kidney; L.kid: Left kidney
DSC: Dice Similarity Coefficient; NSD: Normalized Surface Dice

**Quantitative results**  Table 4 presents the validation results for various organs using the proposed segmentation method. The liver shows the highest performance with a DSC of 93.76% and an NSD of 94.67%, followed closely by the right kidney and left kidney, both of which also demonstrate strong segmentation accuracy with DSC scores of 91.11% and 91.44%, respectively. Organs such as the spleen and aorta also perform well, achieving DSCs above 87%. However, segmentation of smaller or more challenging structures like the right adrenal gland and gallbladder is less accurate, with lower DSCs of 49.78% and 55.86%, respectively, indicating areas for improvement. Overall, the average DSC across all targets is 74.21%, with an average NSD of 80.69%, reflecting the method's robust performance across a variety of organs. The results highlight the method's effectiveness, particularly in segmenting larger, more distinct organs.

**Table 4.** Quantitative evaluation results of the proposed DRL-STNet.

| Target | Validation | |
|---|---|---|
| | DSC(%) | NSD(%) |
| Liver | 93.76 ± 2.74 | 94.67 ± 5.05 |
| Right kidney | 91.11 ± 7.87 | 89.67 ± 9.46 |
| Spleen | 90.26 ± 14.9 | 91.41 ± 16.5 |
| Pancreas | 77.14 ± 13.7 | 89.50 ± 13.3 |
| Aorta | 87.32 ± 8.36 | 92.11 ± 10.9 |
| Inferior vena cava | 74.14 ± 17.6 | 75.99 ± 21.3 |
| Right adrenal gland | 49.78 ± 18.8 | 65.72 ± 25.2 |
| Left adrenal gland | 56.52 ± 19.4 | 69.70 ± 23.1 |
| Gallbladder | 55.86 ± 30.1 | 48.09 ± 29.6 |
| Esophagus | 67.70 ± 14.3 | 85.64 ± 14.2 |
| Stomach | 76.91 ± 14.2 | 80.11 ± 16.7 |
| Duodenum | 52.82 ± 20.4 | 76.70 ± 27.5 |
| Left kidney | 91.44 ± 6.68 | 89.64 ± 7.99 |
| Average | 74.21 ± 16.2 | 80.69 ± 13.8 |

DSC: Dice Similarity Coefficient; NSD: Normalized Surface Dice

**Qualitative results**  Fig. 4 presents segmentation results for various abdominal organs across different cases from the validation set. The first two rows showcase instances where our proposed method achieves accurate segmentation, closely aligning with the ground truth. In these examples, organs such as the liver, kidneys, and spleen are clearly delineated. In contrast, the last two rows illustrate cases where the segmentation is less precise, particularly when comparing the results from our method with and without self-training (ST). These comparisons demonstrate that while our method generally performs well, the inclusion of self-training significantly enhances segmentation quality, especially in challenging scenarios where organs are less distinct or image quality is lower. The segmentation of smaller structures, such as the inferior vena cava, remains

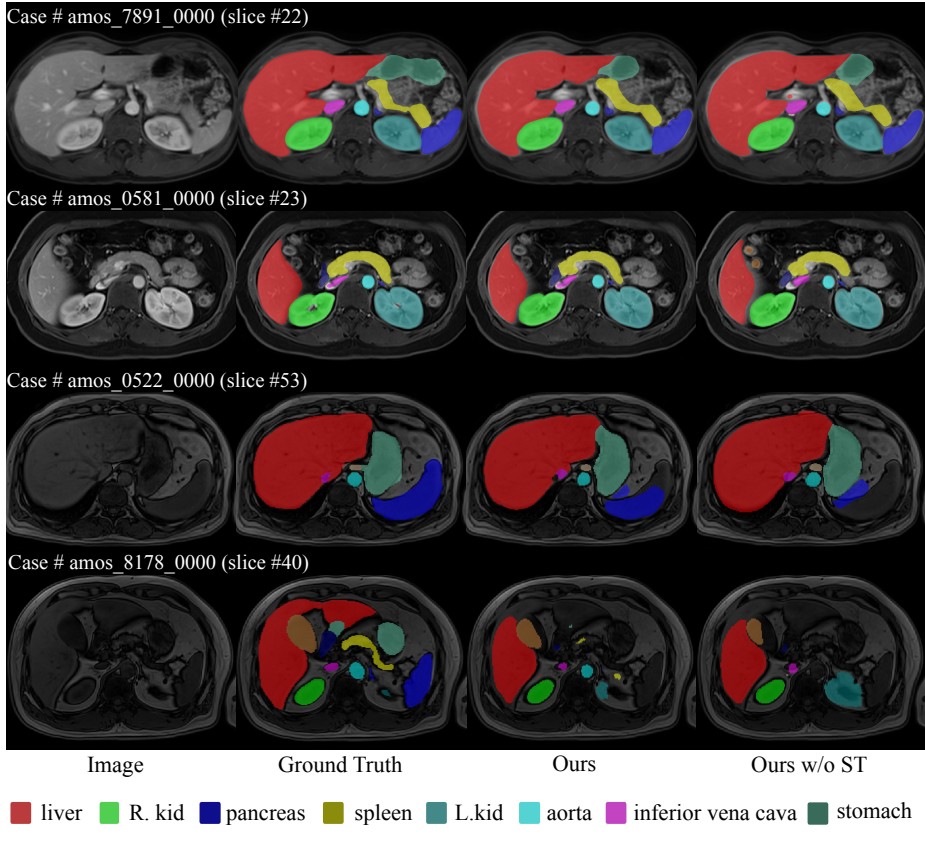

**Fig. 4.** Examples of segmentation results from the validation set. The first two rows illustrate **successful** segmentation outcomes, while the last two rows demonstrate cases with **less accurate segmentation**. The columns represent the original image, ground truth, results from our method, and results from our method without self-training (ST). Different organs are color-coded for clear visualization.

particularly challenging. One direction for future work could be to focus on improving the segmentation accuracy of these smaller and less distinct organs.

**Segmentation efficiency** The segmentation efficiency was quantitatively evaluated by examining the running time and GPU memory consumption across various cases, as detailed in Table 4.2. The experiments were conducted on an NVIDIA Quadro RTX 5000 GPU with 16 GB of memory. The running time varied significantly depending on the image size, with smaller images (e.g., 192 × 192 × 100) requiring as little as 25.21 seconds and larger images (e.g., 1024 × 1024 × 82) taking up to 80.62 seconds. Despite the variation in image sizes and running times, the maximum GPU memory usage remained relatively consistent, hovering around 290 MB to 313 MB across all cases. The total GPU memory consumption, represented as the area under the GPU Memory-Time curve, ranged from 6937 MB to 23075 MB, reflecting the differences in computational demand based on image size and complexity. This analysis highlights the relationship between image size, running time, and GPU memory utilization, emphasizing the scalability of the segmentation process on the selected GPU platform.

**Table 5.** Quantitative evaluation of segmentation efficiency in terms of the running them and GPU memory consumption. Total GPU denotes the area under GPU Memory-Time curve. Evaluation GPU platform: NVIDIA QUADRO RTX5000 (16G).

| Case ID | Image Size | Running Time (s) | Max GPU (MB) | Total GPU (MB) |
|---|---|---|---|---|
| amos_0540 | (192, 192, 100) | 28.49 | 296 | 7992 |
| amos_7324 | (256, 256, 80) | 29.42 | 290 | 8098 |
| amos_0507 | (320, 290, 72) | 31.2 | 313 | 8749 |
| amos_7236 | (400, 400, 115) | 25.36 | 290 | 6995 |
| amos_7799 | (432, 432, 40) | 25.21 | 290 | 6937 |
| amos_0557 | (512, 152, 512) | 84.06 | 290 | 23075 |
| amos_0546 | (576, 468, 72) | 36.54 | 290 | 10072 |
| amos_8082 | (1024, 1024, 82) | 80.62 | 290 | 22146 |

### 4.3   Results on Final Testing Set

Table 6 presents the performance of our model on the testing set in terms of segmentation accuracy and inference efficiency. The average Dice Similarity Coefficient (DSC) and Normalized Surface Dice (NSD) are 48.5% and 50.6%, respectively, with relatively large standard deviations, indicating variation across different cases. Median values of DSC (67.2%) and NSD (69.3%) suggest that in many cases, the model performs reasonably well. The average inference time is approximately 61.9 seconds per volume, with a median of 43.6 seconds, reflecting a moderately consistent runtime across samples.

**Table 6.** Summary of segmentation and computational performance metrics on the testing set.

| Metric | Value |
|---|---|
| DSC mean | $48.5 \pm 32.8$ |
| NSD mean | $50.6 \pm 36.0$ |
| Time mean (s) | $61.9 \pm 56.9$ |
| DSC median | 67.2 |
| NSD median | 69.3 |
| Time median (s) | 43.6 |

DSC: Dice Similarity Coefficient; NSD: Normalized Surface Dice

### 4.4    Limitation and future work

Despite the promising results, our approach has several limitations. First, the method relies heavily on the quality of the disentangled representations and the accuracy of the image translation process. Any errors or inconsistencies in these steps can propagate through the network and affect the final segmentation results. Additionally, while our method works well with the FLARE dataset, its generalizability to other datasets and modalities remains to be thoroughly evaluated.

Another limitation is the potential for synthetic data to introduce artifacts that do not exist in real target modality images. This can lead to segmentation inaccuracies, especially in regions with complex anatomical structures [31]. Furthermore, our approach currently requires significant computational resources and training time, which may limit its practical applicability in real-world clinical settings.

Future research directions can focus on addressing these limitations and improving the robustness and efficiency of the DRL-STNet framework. Potential areas for improvement include:

- **Multi-Modality and Multi-Task Learning:** Extending the framework to handle multiple modalities and tasks simultaneously could improve the generalizability and applicability of the method.
- **Enhanced Representation Learning:** Developing more robust methods for disentangled representation learning to minimize the introduction of artifacts and ensure more accurate image translations. Exploring alternative disentanglement techniques such as variational autoencoders (VAEs) could be beneficial [38].
- **Utilizing Diffusion Models for Domain Transfer:** Investigating the use of diffusion models for domain transfer, which have shown promising results in maintaining high-level semantic information and generating high-quality images [35].
- **Combining Multiple Domain Adaptation Techniques:** Exploring the combination of GAN-based methods with other domain adaptation techniques such as adversarial domain adaptation and self-ensembling methods to enhance robustness and performance [39,3]

## 5   Conclusion

In this paper, we presented DRL-STNet, an innovative framework for unsupervised domain adaptation (UDA) in cross-modality medical image segmentation. By leveraging generative adversarial networks (GANs), disentangled representation learning, and self-training, our method effectively translates images from the source to the target modality, allowing for accurate segmentation of unannotated target images. Experimental results on the FLARE challenge dataset demonstrated that DRL-STNet outperforms state-of-the-art methods in both the Dice similarity coefficient and Normalized Surface Dice metrics, particularly in segmenting abdominal organs.

In summary, while DRL-STNet shows great potential for unsupervised domain adaptation in medical image segmentation, there are several areas where further research and development are needed to enhance its performance and applicability. Addressing these challenges will be crucial for the successful integration of UDA techniques in clinical practice.

**Acknowledgements** We have not used any pre-trained models or additional datasets other than those provided by the organizers. The proposed solution is fully automatic without any manual intervention. We thank all data owners for making the CT scans publicly available and CodaLab [40] for hosting the challenge platform.

## Disclosure of Interests

The authors declare no competing interests.

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

**Table 7.** Checklist Table. Please fill out this checklist table in the answer column.

| Requirements | Answer |
| --- | --- |
| A meaningful title | Yes |
| The number of authors ($\leq 6$) | 6 |
| Author affiliations and ORCID | Yes (no ORCID) |
| Corresponding author email is presented | Yes |
| Validation scores are presented in the abstract | No |
| Introduction includes at least three parts: background, related work, and motivation | Yes |
| A pipeline/network figure is provided | Fig. 1 |
| Pre-processing | Section 3.1 |
| Strategies to use the partial label | Section 3.1 |
| Strategies to use the unlabeled images. | Section 3.1 |
| Strategies to improve model inference | None |
| Post-processing | None |
| The dataset and evaluation metric section are presented | Table 4 |
| Environment setting table is provided | Table 1 |
| Training protocol table is provided | Table 2 |
| Ablation study | Table 3 |
| Efficiency evaluation results are provided | Table 5 |
| Visualized segmentation example is provided | Fig. 4 |
| Limitation and future work are presented | Yes |
| Reference format is consistent. | Yes |