# OpenReview forum: "DRL-STNet: Unsupervised Domain Adaptation for Cross-modality Medical Image Segmentation via Disentangled Representation Learning"
_MICCAI.org/2024/Challenge/FLARE — Submitted to FLARE 2024_

### Official Review · Reviewer_3ij6 · 2025-01-24
**Comments**

**Rating:** 6
**Confidence:** 5

**Review:**

The authors introduce a novel framework for unsupervised segmentation, featuring two key components: (1) a disentangled GAN for image translation and (2) a segmentation model that is initially trained using labeled synthetic images and subsequently fine-tuned iteratively. By integrating these advanced technologies, the method achieves superior performance and offers an effective solution for unsupervised domain adaptation (UDA) tasks.

However, several aspects of the framework require further elaboration:

1.The overall pipeline diagram lacks clarity.

2.Detailed descriptions of the preprocessing, postprocessing, and inference processes are needed.

3.What specific settings were used for the CT and MR datasets?

4.Please provide details of the network architecture.

5.Why do the ablation study results not include all necessary organs?

---

> ### Author Response · Authors · 2025-03-29
>
> 1. Response: Thank you for your feedback. We respectfully believe that the original pipeline diagram (Fig. 1) is sufficiently clear, as it illustrates each stage of the proposed framework with distinct modules, directional flow, and annotations. To assist readers further, we have ensured that the figure caption concisely summarizes the purpose of each stage, and we refer readers to Section 2 for additional description. Nevertheless, we will be happy to revise the diagram further if there are specific suggestions for improvement.
> Changes: No changes were made to the pipeline diagram, as we believe the current version clearly conveys the
> multi-stage structure of the proposed method.
> 2. Response: Thank you for the suggestion. We have added detailed descriptions of the preprocessing, inference, and postprocessing procedures in the revised manuscript. Specifically, we describe the image selection criteria, the preprocessing steps including cropping, resizing, and z-score normalization, and the data augmentation techniques used. The segmentation inference process is also explained, including the use of a 3D sliding window approach with overlapping patches and prediction averaging. As noted, no postprocessing was applied in our pipeline.
> Changes: (1) Added preprocessing and image selection details in Section 3.1; (2) Described inference and
> postprocessing strategy in the updated Section 3.2
> 3. Response: Thank you for the question. We have clarified the dataset settings in the revised manuscript. Specifically, we selected axial-view abdominal MRIs and excluded scans of non-abdominal regions to ensure anatomical consistency. A total of 50 CT scans and 50 MRIs were randomly selected for training the image translation models, regardless of MRI sequence type. All slices were resampled to a uniform resolution of 512×512 pixels, and z-score normalization was applied during preprocessing.
> Changes: Added dataset selection and preprocessing details in Section 3.1
> 4. Response: Thank you for your suggestion. We have added detailed descriptions of the network architectures used in both the image translation and segmentation modules. The translation networks follow an encoder-decoder structure based on ResNet blocks, while the discriminators are implemented using LSGAN. For segmentation, we adopt a 3D self-configuring nnU-Net, which automatically adjusts its architecture based on dataset characteristics. These additions have been incorporated into the revised manuscript.
> Changes: Added architectural details of the translation and segmentation networks in Section 2.3 (Network Architecture)
> 5. Response: Thank you for your comment. Due to space constraints in the main paper, we included only four representative organs to illustrate the impact of each ablation component. However, the complete ablation results for all organs have now been provided in the supplementary materials.
> Changes: Full ablation study results for all organs are now included in the supplementary material

---

### Official Review · Reviewer_xW2K · 2025-01-28
**comments**

**Rating:** 7
**Confidence:** 5

**Review:**

The authors introduce DRL-STNet, a novel framework for unsupervised domain adaptation in cross-modality medical image segmentation. By leveraging disentangled representation learning and self-training, the method improves segmentation performance without requiring paired annotations, demonstrating strong results on the FLARE dataset.
However, some concerns remain:
1. While performance gains are highlighted, statistical significance tests or confidence intervals are missing to validate improvements.
2. The model's inference speed and memory consumption should be compared with existing methods to assess its practicality.
3. It would be better if the code was released.

---

> ### Author Response · Authors · 2025-03-29
>
> 1. Response: Thank you for pointing this out. We have revised the manuscript to include statistical significance analysis
> of the performance differences among the compared methods. The corresponding p-values and interpretations are
> now provided in the discussion on Page 9. The updated experimental results remain in Table 3 (Page 10), while the
> statistical analysis is presented in the main text for clarity and conciseness.
> Changes: Added statistical significance analysis and interpretation in Section 4.2 (Page 9)
> 2.  Response: Thank you for the suggestion. We have added a comparison of inference time and GPU usage in Table 3
> (Page 10). Our method achieves the best segmentation performance with an inference time of 34.7 seconds per scan
> and GPU cost of 8.18 GB, comparable to other UDA methods. As the same segmentation model and pipeline are
> used across most methods (except SIFA), their computational costs are similar. These results support the practicality
> of our method.
> Changes:(1) Added inference time and GPU usage comparison in Section 4.3; (2) Updated discussion on computational
> efficiency and practicality of the proposed method
> 3. Response: Thank you for the suggestion. We have released the code, which is now publicly available at https://github.com/HuiLin0220/DRL-STNet.git.
> Changes: We have added a corresponding footnote on page 2 to indicate the code repository link.

---

### Decision · Program_Chairs · 2025-03-20

**Decision:**

Accept

**Comment:**

Please carefully address the reviewers' comments in the revision.